# A Fluorescent and Magnetic Hybrid Tracer for Improved Sentinel Lymphadenectomy in Prostate Cancer Patients

**DOI:** 10.3390/biomedicines11102779

**Published:** 2023-10-13

**Authors:** Svenja Engels, Bianca Michalik, Lena Dirks, Matthias N. van Oosterom, Friedhelm Wawroschek, Alexander Winter

**Affiliations:** 1University Hospital for Urology, Klinikum Oldenburg, Department of Human Medicine, School of Medicine and Health Sciences, Carl von Ossietzky University Oldenburg, 26129 Oldenburg, Germany; engels.svenja@klinikum-oldenburg.de (S.E.); michalik.bianca@klinikum-oldenburg.de (B.M.); lena.dirks@uni-oldenburg.de (L.D.); wawroschek.friedhelm@klinikum-oldenburg.de (F.W.); 2Interventional Molecular Imaging Laboratory, Department of Radiology, Leiden University Medical Center, 2300 RC Leiden, The Netherlands; m.n.van_oosterom@lumc.nl

**Keywords:** hybrid, fluorescence, magnetic tracer, sentinel lymph node, prostate cancer

## Abstract

In prostate cancer, sentinel lymph node dissection (sLND) offers a personalized procedure with staging ability which is at least equivalent to extended LND while inducing lower morbidity. A bimodal fluorescent–radioactive approach was introduced to improve sentinel LN (SLN) detection. We present the first in-human case series on exploring the use of a fluorescent–magnetic hybrid tracer in a radiation-free sLND procedure. Superparamagnetic iron oxide nanoparticles and indocyanine green were administered simultaneously in five prostate cancer patients scheduled for extended LND, sLND and radical prostatectomy. In situ and ex vivo fluorescence and magnetic signals were documented for each LN sample detected via a laparoscopic fluorescence imaging and magnetometer system. Fluorescence and magnetic activity could be detected in all patients. Overall, 19 lymph node spots could be detected in situ, 14 of which were fluorescently active and 18 of which were magnetically active. In two patients, no fluorescent LNs could be detected in situ. The separation of the LN samples resulted in a total number of 30 SLNs resected. Ex vivo measurements confirmed fluorescence in all but two magnetically active SLNs. One LN detected in situ with both modalities was subsequently shown to contain a metastasis. This study provides the first promising results of a bimodal, radiation-free sLND, combining the advantages of both the magnetic and fluorescence approaches.

## 1. Introduction

In prostate cancer, in addition to the relevance of the prostate-specific antigen (PSA) value for diagnosis [1], the lymph node (LN) status is a relevant prognostic factor for the oncological outcome, as well as for planning adjuvant therapy [2]. LN metastasis occurs in 3–80% of prostate cancer patients undergoing radical prostatectomy [3]. Despite the rise of new imaging techniques, such as prostate-specific membrane antigen (PSMA) PET/CT [4,5], extended lymph node dissection (eLND) remains the gold standard for LN staging in clinically localized prostate cancer [6]. However, while the invasiveness and the risk of associated complications and/or morbidity increase with the number of LNs removed [2,7], the prevalence of LN involvement directly correlates with the number of dissected LNs and/or the extent of LND [8,9]. Furthermore, there are indications that the dissection of LN metastases as part of pelvic LND (PLND) has therapeutic benefit, especially for patients with minimal lymph node invasion (LNI) [10,11,12]. In this context, sentinel lymph node (SLN) dissection (sLND) was established in prostate cancer to reduce the number of dissected LNs without a loss of sensitivity as it enables the surgeon to selectively remove the LNs which drain directly from the prostate and thus bear the greatest risk of harboring metastasis [13,14,15].

There is accumulating evidence that a fluorescent–radioactive hybrid tracer outperforms other conventional tracers in its nodal staging ability [16,17]. Furthermore, using a fluorescent–radioactive hybrid tracer during sLND to label SLNs may be associated with lower rates of biochemical recurrence than eLND and with low rates of complications as well [18,19]. This bimodal approach for the detection of SLNs seems especially advantageous for minimally invasive and robot-assisted surgery [20], but it has the disadvantages of depending on the availability of nuclear medicine and exposing patients and surgical staff to radiation. Magnetic tracers consisting of superparamagnetic iron oxide nanoparticles (SPIONs) were introduced as radiation-free alternatives to conventional radiotracers in various tumor entities including breast cancer and prostate cancer [21,22]. In a recent analysis of data from 3000 patients, our group showed equally high levels of reliability of both the radiotracer and the magnetic tracer in detecting LN metastasis [23]. Unfortunately, laparoscopic magnetometer probes for the SPION-labelled detection of SLNs are currently being developed but are not yet approved for routine clinical use [24,25,26]. Analogous to the fluorescent–radioactive hybrid tracer, additional fluorescence labeling of the SPION tracer could provide a further benefit for SLN detection, especially in minimally invasive robotic surgery. To our knowledge, there are only a few pre-clinical studies with fluorescent–magnetic hybrid tracers which show the general feasibility of fluorescent–magnetic SLN detection in animal models and human skin tissue explants [25,27,28,29,30]. In this small case series (according to IDEAL Stage 2a [31]), we therefore introduce the first human in situ implementation of a fluorescent–magnetic hybrid tracer for radiation-free bimodal sLND during radical prostatectomy.

## 2. Materials and Methods

### 2.1. Patients

This case series included five prostate cancer patients who were scheduled for a radical prostatectomy in combination with both extended pelvic LND and sLND at the University Hospital for Urology Oldenburg, Germany, in 07/2022. The patients did not undergo any pre-treatment that might have altered the lymphatic drainage of the prostate, such as anti-hormonal therapy or transurethral prostate surgery. All patients were informed verbally and in writing about radical prostatectomy, magnetometer-guided sLND and the use of indocyanine green (ICG) for LN visualization. All patients signed a consent form regarding the use of their clinical data in a scientific publication.

### 2.2. Tracer

We used a commercially available magnetic tracer (Magtrace, Endomagnetics Ltd., Cambridge, UK) consisting of superparamagnetic iron oxide nanoparticles (SPIONs) coated with carbodextran and dissolved in saline. To synthesize the fluorescent–magnetic hybrid tracer, either 50 µL or 100 µL of 5 mg/mL ICG (Verdye, Diagnostic Green GmbH, Aschheim-Dornach, Germany; dissolved in water for injection) was added to 2 mL of the SPION tracer, similar to the procedure described before [27]. To test for the effects of concentration on fluorescence detection in situ, we applied 50 µL of the fluorescent tracer (i.e., a total amount of 0.25 mg of ICG) in the first three patients and raised this to 100 µL (i.e., a total amount of 0.5 mg of ICG) in the other two patients.

### 2.3. Surgery

The day before surgery (median 19.3 h, IQR 18.6–21.6 h), a urologist transrectally injected the fluorescent–magnetic hybrid tracer under ultrasound guidance into each lobe of the prostate in three deposits, respectively [22]. PLND was performed during open retropubic surgery. The lymphatic drainage area of the prostate was searched for fluorescent LNs via a near-infrared camera (IMAGE1 S™ Rubina^®^, Karl Storz SE & Co. KG, Tuttlingen, Germany) and for magnetic LNs via a handheld magnetometer probe (Sentimag^®^, Endomagnetics Ltd., Cambridge, UK). All LN spots identified in situ via either mode were selectively surgically removed. As recommended by the European guidelines on prostate cancer [6], PLND was completed by bilateral eLND according to the standard anatomical template. The lymphatic tissue samples from the LN spots detected in situ LN spots could frequently be separated into several single LNs ex vivo, resulting in a higher number of SLNs measured ex vivo. All LNs as well as the radical prostatectomy specimens were checked for ex vivo fluorescence and magnetic activity, respectively. After surgery, all tissue samples were fixed in formalin for approximately 24 h and were then routinely processed. For each tissue sample, hematoxylin–eosin stains were microscopically analyzed for tumor infiltration by a pathologist experienced in uropathology.

## 3. Results

The clinical and histopathological characteristics of the five patients and the details of the tracer injection and in situ detection data are summarized in Table 1.

In all five patients, we could detect magnetic activity and fluorescence. In the prostates of four of the five patients, we could detect magnetic activity as well as fluorescence (Table 1). In patient #2, we were not able to measure the fluorescence of the prostate in situ as the fluorescence camera was not ready to use at this time during the surgery. LN samples could be located in all patients via magnetic activity and/or fluorescence (Table 1). Figure 1 shows the in situ and ex vivo fluorescence camera images of the prostate, as well as of one LN of patient #4.

All histopathologically confirmed LNs detected via fluorescence were also magnetically active. One fluorescent-only LN sample turned out to be lymphatic fatty tissue upon histopathological examination (indicated by ** in Table 1). In two of the three patients with the lower concentrations of ICG, we could only detect magnetically labeled but non-fluorescent LNs in situ (patients #2 and #3, Table 1). In one of these two patients, ex vivo measurements showed fluorescence in all magnetically active LNs as well (patient #3, Table 1). In the other patients, two of the SLNs were magnetically active only (patient #2, Table 1). It should be noted that the number of detected SLN samples in situ (*n* = 19) is not equivalent to the total number of SLNs identified via histopathology (*n* = 30) (Table 1), as a detailed ex vivo investigation may reveal several single LNs in one dissected LN tissue sample. Overall, there was a high level of accordance between the two modalities, with 28 of 30 SLNs being both magnetically active and fluorescent and the remaining two found to be only magnetically active. A histopathological examination revealed the metastasis of one LN in an SLN of one patient (patient #1, Table 1). The metastatic SLN was detected in situ via both detection modalities. An examination of the 20 additional LNs removed during ePLND revealed no further metastases.

## 4. Discussion

This study represents the first in-human description of the in situ administration of an ICG-SPION hybrid tracer for LN visualization in prostate cancer patients. Our data show the principal applicability of this hybrid tracer that allows for the magnetic and fluorescent in situ identification of SLNs draining directly from the prostate.

Applying the ICG-SPION hybrid tracer in prostate cancer patients, we observed magnetic activity in SLNs comparable to use of the traditional SPION tracer only. Despite the very small sample size in our study, the number of SLNs, the overall number of LNs removed and the number of LN metastases are comparable to prior studies and data collected in our center using the magnetic tracer alone [23]. Therefore, the addition of ICG does not seem to impair the magnetic tracer or influence the lymphatic drainage or accumulation of the magnetic tracer within the LNs. This is also in line with the results of animal experiments using different SPION-ICG applications [25,27,28,29,30].

ICG alone has been used as fluorescent lymphangiographic agent, labeling LNs in various tumor applications such as prostate cancer [32], breast cancer [33], uterine cancer [34], cervical cancer [35] and gastric cancer [36]. Since it is non-toxic at doses below 2 mg/kg body weight, has a low rate of adverse reactions [37] and is highly reliable, ICG has been recommended for SLN visualization, e.g., in endometrial cancer, by several international guidelines [38]. While the intraoperative use of ICG provides the advantage of tracking lymphatic pathways in real time, it is not specific to SLNs [39], and its limited penetration depth (<1.5 cm [40]) makes it a superficial technique, lacking an option for preoperative imaging. As such, relying on fluorescence imaging only might complicate the identification of SLNs depending on their anatomical location [41].

In prostate cancer, a fluorescent–radioactive hybrid tracer [16] therefore seems to be a suitable alternative for SLN tracking [17]. The larger molecular size of the tracer complex compared to “free” ICG slows down the drain and leads to greater accumulation within the SLNs (i.e., increased SLN specificity), allowing for tracer injection to be carried out one day preoperatively and the preoperative localization of the SLNs via imaging, which enhances intraoperative detection. This bimodal approach combines the advantages of the approved radio-guided SLN procedure, including preoperative lymphatic mapping with an intraoperative radioactive and/or fluorescence confirmation of the localization of the SLNs [20]. In our center, we use a magnetic SPION tracer as a reliable and radiation-free alternative to radioisotope SLN labeling. It was first introduced in breast cancer [21,42,43] and later on in several other tumor entities, including prostate and penile cancer [22,23,44]. This technique also allows for the pre-operative mapping of magnetically labelled SLNs through magnetic resonance imaging comparable to scintigraphy in radioactive marking [45]. As an additional advantage, the fluorescent signal also allows for the visual identification of SLNs, e.g., close to the prostate, where it is not possible to differentiate between the magnetic signals of the prostate and the SLNs. While the use of pure ICG for LND might cause the problem of leakage into the surgical field [46], this seems not to be the case when using hybrid tracers like our fluorescent–magnetic tracer or an ICG-99mTc-nanocolloid [17] for LNDs, probably because of the larger molecular size.

The development of a fluorescent–magnetic hybrid tracer is therefore the next logical step toward optimizing the detection of SLNs, especially for minimally invasive and robotic PLND during radical prostatectomy. After our own successful animal studies in which we combined the magnetic tracer that we normally use in our clinical routine with ICG, its use in this small case series was then required to achieve this purpose.

## 5. Conclusions

Our study presents the first promising results of radiation-free bimodal SLN visualization in prostate cancer patients. Future studies need to determine the optimal concentration of ICG within the SPION tracer and test the reliability of the fluorescent–magnetic hybrid tracer with larger numbers of patients, including preoperative visualization via MRI.

## Figures and Tables

**Figure 1 biomedicines-11-02779-f001:**
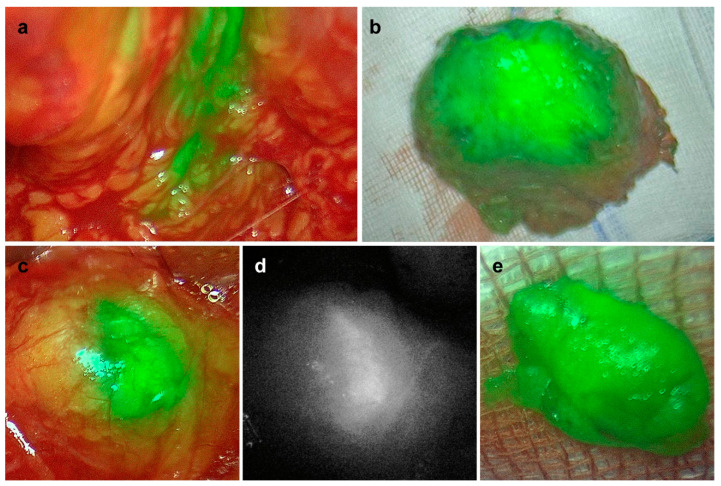
In vivo (**a**,**c**,**d**) and ex vivo (**b**,**e**) detection of fluorescence in the prostate (**a**,**b**) and in one lymph node (**c**–**e**) of patient #4. (**a**–**c**,**e**) Overlay; (**d**) monochromatic.

**Table 1 biomedicines-11-02779-t001:** Clinical and histopathological patient characteristics and in situ detection data.

Patient	#1	#2	#3	#4	#5	Overall
Clinical information						
Age [years]	62	70	69	67	70	
BMI	24	24	22	27	23	
Total PSA [µg/L] ^†^	4.4	2.8	19.1	2.3	5.9	
Clinical tumor category	2	2	3	1c	2	
Biopsy ISUP grade	4	4	3	1	2	
Positive biopsy cores [%]	7	58	29	17	21	
Risk of LNI ^††^ [%]	34	63	64	3	12	
Tracer injection						
Total amount of ICG [mg]	0.25	0.25	0.25	0.5	0.5	
Time injection to surgery [h]	18.6	22.4	19.3	16.8	21.6	
In situ detection						
Prostate magnetic activity	yes	yes	yes	yes	yes	
Prostate fluorescence	yes	n.a. *	yes	yes	yes	
No. of lymph node spots	3	2	2	7	5	19 **
No. of magnetically active spots	3	2	2	7	4	18
No. of fluorescent spots	2	0	0	7	5 **	14
Ex vivo detection						
Prostate magnetic activity	yes	yes	yes	yes	yes	
Prostate fluorescence	yes	yes	yes	yes	yes	
No. of LNs resected	13	8	12	8	9	50
No. of SLNs resected	5	6	5	7	7	30
No. of magnetically active SLNs	5	6	5	7	7	30
No. of fluorescent SLNs	5	4	5	7	7 **	28
Post-operative information						
Pathological tumor category	2c	3b	2c	2c	3a	
Post-operative ISUP grade	5	5	3	2	2	
pN	1	0	0	0	0	

^†^ Total PSA values were determined in our laboratory using electrochemiluminescence immunoassay (Elecsys^®^ total PSA, Roche Diagnostics GmbH, Mannheim, Germany) using a cobas^®^ e801 analytical unit (Roche Diagnostics GmbH, Mannheim, Germany); ^††^ as calculated from our nomogram [3]; * no in situ measurement as the fluorescence camera was not yet ready to use; ** one fluorescent lymph node sample turned out to be lymphatic fatty tissue upon histopathological examination. BMI: body mass index; ICG: indocyanine green; ISUP: International Society of Urological Pathology; LNI: lymph node involvement; *n*: number; pN: pathological nodal status; PSA: prostate-specific antigen; SLNs: sentinel lymph nodes.

## Data Availability

The data presented in this study are available upon request from the corresponding author.

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
