# Peer review of "A Fluorescent and Magnetic Hybrid Tracer for Improved Sentinel Lymphadenectomy in Prostate Cancer Patients"

_biomedicines, 2023, doi:10.3390/biomedicines11102779_

Round 1

Reviewer 1 Report

Abstract

Lines 22-23: how many LNs were removed in total? Or at least add the percentage to the number of fluorescent or magnetic LNs. Median 2-3 per patient? Please clarify.

Line 26: again, it is unclear what that median refers to.

Line 26-27: both modalities detected the metastasis. How?

Introduction

Line 35: it is okay to use directly PET/CT, but I would rather write PSMA in the extended form the first time.

Line 40: does PLND P stand for Prostatic? Please write extended the first time.

Lines 47-48: sLND already specified use the acronym directly.

Results

Lines 133-135: (considering materials and methods lines 96-97) If I get correctly, you identified a median of in situ 3 magnetic and 2 fluorescent LNs and ex vivo a median of 6 magnetic and 5 fluorescent SLNs. And you removed a total of 47 SLNs from all 5 patients. Only 1 / 47 SLNs were positive. But again, correct me if I did not get it right; you stated you performed eLND/PLND. How many LNs were removed with the latter procedure that were not sentinel? How many of them, if any, were metastatic? I assume none. Anyway, I think it would be interesting to know the total number removed.

Discussion

Lines 146-147: I would also add Kuwahata et al. 2018 Int J Nanomedicine 13:2427-2433. doi: 10.2147/IJN.S153163. if the authors think that what they used is similar to the Sienna+ / ICG association

In line with my comment on the results section about a more accurate resume on the number of LNs removed / SLNs / metastatic LNs, I would like to have the Authors comment and discuss some more about it.

Reviewer 2 Report

The authors describe results of a bimodal, radiation-free sLND with staging ability for Prostate cancer(PCa).

1.0 Epidemiological data about PCa.

May this  technique  be applied to overall Pca or only to advanced PCa?

This is not clear to me.

2.0 PSA is the reference biomarker but in the introduction there is no information about its application, and it has a great relevance for prediction and prognosis. I recommend to consider in the introduction the following updated paper for reporting information about its use, since PSA values are obviously reported in this paper.

Ferraro S, Biganzoli D, Rossi RS, Palmisano F, Bussetti M, Verzotti E, Gregori A, Bianchi F, Maggioni M, Ceriotti F, Cereda C, Zuccotti G, Kavsak P, Plebani M, Marano G, Biganzoli EM.Individual risk prediction of high grade prostate cancer based on the combination between total prostate-specific antigen (PSA) and free to total PSA ratio.

Clin Chem Lab Med. 2023 

3.0 Consider to report information about PSA assay or the estimate of mean/median values is biased.

4.0 Explain the reason why you have not considered to ask for IRB approval, informed consent is not enough.

Round 2

Reviewer 2 Report

The authors has answered: "We agree that PSA has great relevance for screening, prediction, and prognosis. Nevertheless, in our study the reported PSA values together with clinical tumor category and ISUP grade (and percentage of positive cores) are only used to indicate the progress of the tumor (EAU risk group) or risk of lymph node invasion according to our nomogram, respectively."

As reviewer I agree that the role of PSA for indicating the tumor progress/risk of lynphnode is relevant. And the reference I have previously recommended indicates PSA values in relation to ISUP grade risk. So I think that some information has to be reported in the introduction in agreement with the reference Ferraro S, et al. Individual risk prediction of high grade prostate cancer based on the combination between total prostate-specific antigen (PSA) and free to total PSA ratio. Clin Chem Lab Med. 2023.

In addition I have asked to report the information on PSA assay. In the same table you have reported the data which cannot be compared if obtained by different assays  

Round 3

Reviewer 2 Report

no further comments